# The elimination of trachoma as a public health problem in Togo: Successes and challenges

**Akila Wimima Bakoubayi**[1,2]*, **Denis Agbenyigan Yawovi Gadah**[1], **Piham Gnossike**[3], **Wendpouiré Ida C. Zida-Compaore**[4], **P'tanam P'kontème Bando**[1], **Kamevor Alaglo**[3], **Maweke Tchalim**[3], **P'niwè Patchali**[3], **Alexandra Bitty-Anderson**[5], **Didier Koumavi Ekouevi**[2,4]

1 German Leprosy and Tuberculosis Relief Association, Lomé, Togo, 2 University of Lomé, Faculty of Health Sciences, Department of Public Health, Lomé, Togo, 3 National Programme for Neglected Tropical Diseases, Lomé, Togo, 4 African Centre for Research in Epidemiology and Public Health, Lomé, Togo, 5 PACCI Program Ivory Coast site, Abidjan, Côte d'Ivoire

* bakoubayiakila@gmail.com

**Data Availability Statement:** Data are available upon request from the National Neglected Tropical Disease Program of Togo. These data belong to the Togolese Ministry of Health of which the program

## Abstract

### Background

As of May 2022, 15 countries have declared that they have reached their trachoma elimination targets, but only 13 of them, including Togo, have been validated by the World Health Organization as having eliminated the disease as a public health problem. The aim of this study was to describe the broad interventions that have supported the elimination of trachoma as a public health problem in Togo from its inception in 2006 to the validation of its elimination in 2022.

### Method

A review and compilation of data and information contained in the country's submission to World Health Organization for validation of trachoma elimination as a public health problem was conducted. Data from national and local surveillance systems and reports on actions taken after achieving the elimination target were also included.

### Results

Togo has achieved the elimination of trachoma as a public health problem by 2022. The prevalence of follicular trachoma among children aged 1–9 years is <5% in all nationally defined administrative units suspected of having trachoma after stopping mass treatment for at least 2 years. The prevalence of trichiasis among persons aged 15 years and older is less than 0.2% in all administrative units previously endemic for trachoma and evidence of the ability to manage incident cases of emerging trichiasis in the community has been demonstrated. The key of the success in the elimination process was primarily the political commitment of the health authorities with financial and technical support from various international organizations.

### Conclusion

The elimination of trachoma as a public health problem in Togo is a real success story that can serve as an example for the elimination of other neglected tropical diseases in Africa.

is a part. Data requests can be sent to the Togolese Ministry of Health at pnmtntogo@gmail.com.

**Funding:** The author(s) received no specific funding for this work.

**Competing interests:** The authors have declared that no competing interests exist.

But regular monitoring and surveillance is essential to avoid the re-emergence of such disease in the country.

## Author summary

Trachoma remains a public health problem in 43 countries in 2022, including 26 countries in the African region. During the 72nd Session of the World Health Organization Regional Committee for Africa held in Lomé, Togo, from 22 to 26 August 2022, World Health Organization has honored Togo for becoming the first country in the world to eliminate four neglected tropical diseases in only 11 years namely dracunculiasis (Guinea worm), lymphatic filariasis, human African trypanosomiasis (sleeping sickness) and trachoma, the last of which was eliminated in May 2022. Between 2006 and 2016, different surveys were conducted to identify trachoma endemic areas, scree n and manage Trachomatous trichiasis cases. Final epidemiological surveys conducted in 2017 confirmed that there was sufficient evidence to show that elimination should be achieved in all endemic areas prompting the formal submission of the country's elimination dossier to the World Health Organization.

By reaching this milestone, Togo has improved the quality of life of people living in former trachoma endemic areas by avoiding visual impairment and eventual blindness due to this disease. Post- validation monitoring and surveillance is essential to detect the resurgence of the disease because elimination as a public health problem is a reversible state.

## 1. Introduction

Trachoma is the leading infectious cause of blindness in the world [1]. Caused by *Chlamydia trachomatis*, the infection can lead to trachomatous follicular inflammation, a clinical sign used to estimate the presence of active trachoma [2]. Repeated infections lead to the development of scarring on the tarsal conjunctiva of the upper eyelid which, over time, can cause upper eyelid deformity [2]. Deformed upper eyelids may result in trichiasis (the presence of eyelashes touching the front of the eye), the low prevalence of which is a criterion for eliminating trachoma. [3].

The World Health Organization (WHO) Alliance for the Global Elimination of Trachoma had called for the elimination of trachoma as a public health problem by 2020 [1]. However, trachoma remains a public health problem in 43 countries in 2022, including 26 countries in the African region [4]. As of May 2022, 15 countries have been declared reaching their trachoma elimination targets, but only thirteen (13) of them including Togo had been certified by WHO as having eliminated the disease as a public health problem [5].

During the 72nd Session of the WHO Regional Committee for Africa held in Lomé, Togo, from 22 to 26 August 2022, WHO honored Togo for becoming the first country in the world to eliminate four neglected tropical diseases in only 11 years namely dracunculiasis (Guinea worm), lymphatic filariasis, human African trypanosomiasis (sleeping sickness) and trachoma, the last of which was eliminated in May 2022 [4]. Togo thus becomes the five African country to be certified by WHO as having reached the milestone of trachoma elimination after Morocco in 2016, Ghana in 2018, Gambia in 2021 and Malawi in 2022 [4,5].

This article reviews the history of trachoma control activities and summarizes the extensive interventions of the Ministry of Health, Public Hygiene and Universal Access to Health Care

through the National Programme for Neglected Tropical Diseases to eliminate this neglected tropical disease (NTD) in Togo from its launch in 2006 until the validation of its elimination in 2022.

## 2. Method

### 2.1 Ethics statement

This study used anonymized case data, which were collected by the National Neglected Tropical Disease Program as part of its program surveillance activities. In this study, no interventions were performed (neither diagnostic, nor therapeutic, nor survey). We relied solely on a retrospective collection of anonymous cases authorized by the program. We have not been in contact with any individual. We do not require consent and could not obtain it in this context.

### 2.2 Geographic setting

Togo is a West African country, located on the southern coast of West Africa with an area of 56,600 sq. km. It borders Burkina Faso in the north, the Atlantic Ocean in the south, Benin in the east and Ghana in the west (Fig 1). Its current population, estimated at 7.88 million inhabitants, is unevenly distributed over the country, with about 24% concentrated in the capital, Lomé [6]. For health-related interventions, Togo has 6 regions and 39 health districts.

### 2.3 Activities and interventions to fight trachoma in Togo

The organization of trachoma control began within the National Program for Blindness control created in 1989 by the Ministry of Health.

Between 2006 and 2016, different surveys were conducted to identify trachoma endemic areas, screen and manage Trachomatous trichiasis (TT) cases.

**2006 surveys.** In 2006, a school-based survey was conducted among children aged 6–9 years in the Binah district (in the Kara region in the north of the country). The data were collected using "integrated threshold mapping", a canevas designed by CDC Atlanta. Data were collected simultaneously on trachoma, schistosomiasis and geo-helminthiasis in order to identify villages in the Binah district where the prevalence of these three diseases exceeded the threshold for initiating mass treatment according to WHO guidelines.

**2009 Surveys.** In 2009, a survey was conducted in both school and community facilities in 14 of the 15 health districts in the three northern regions of Togo (Centrale, Kara and Savanes) with the exception of the Binah district (where a similar survey had already been conducted in 2006). This survey also used the "integrated threshold mapping" approach for children aged 1 to 6 years.

There was no TT case in these first two surveys in 2006 and 2009.

**2011 Survey.** In 2011, a community cluster survey was conducted in the three districts (Blitta, Sotouboua and Binah) where trachomatous inflammation—follicular (TF) appeared to be prevalent at levels above the elimination threshold in the 2006 and 2009 surveys. It estimated the prevalence of TF among children aged 1–9 years and women aged 15 years and older in these three districts.

**Active and passive screening and TT surgery from 2015 to 2016.** A TT screening and surgery project were run in 7 districts in the Kara region and 4 districts in the Central region from June 2015 to December 2016. These two regions were chosen based on the results of the 2009 and 2011 surveys. Ten trained head nurses per district identified suspected TT cases that were confirmed by senior ophthalmic technicians. Confirmed cases were operated on by the senior ophthalmic technicians under the supervision of an ophthalmic surgeon.

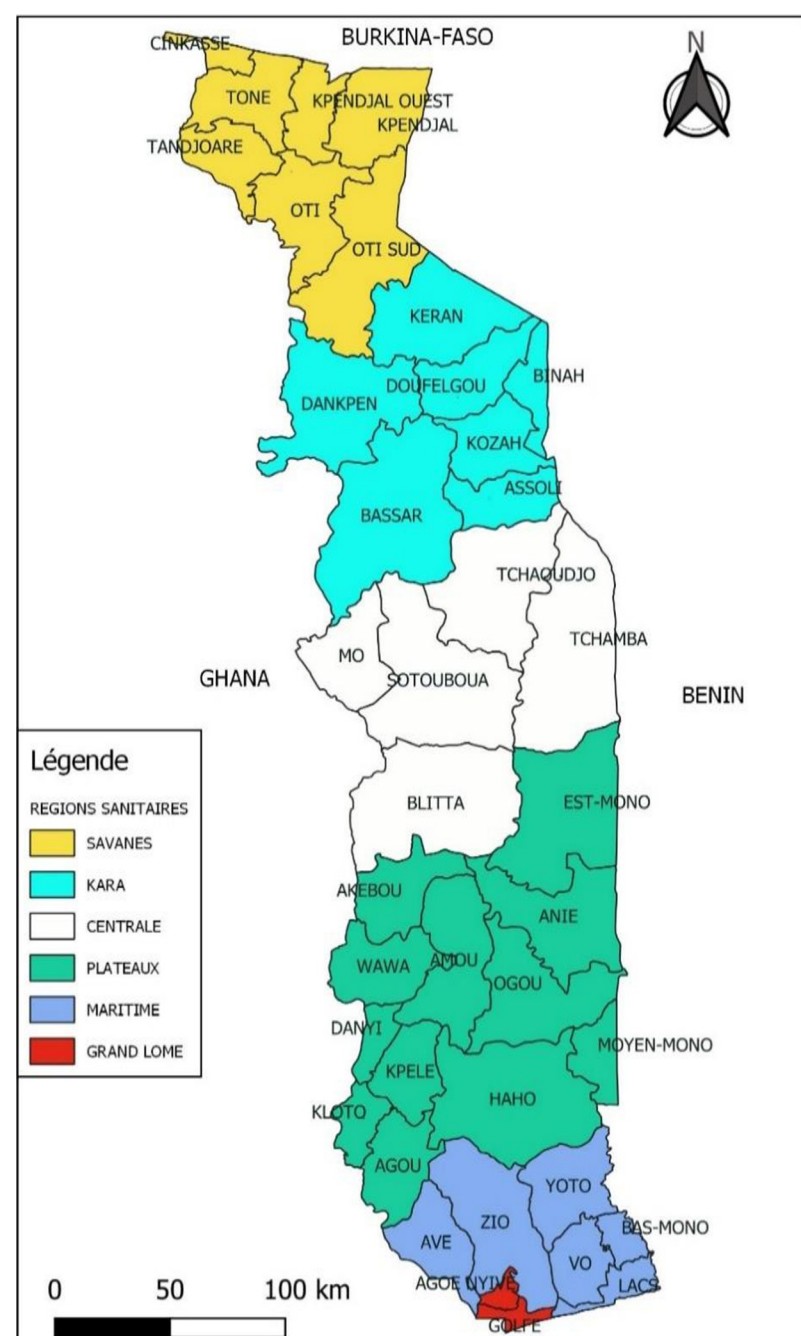

**Fig 1. Regions and health districts of Togo.** Made with Natural Earth, http://www.naturalearthdata.com/about/terms-of-use

**Detection of suspected TT cases during mass treatment from 2015 to 2017.**   Since 2015, the NPNTD has been organizing the Integrated mass treatment with Ivermectin, Albendazole and Praziquantel against onchocerciasis, urinary bilharzia and geo-helminthiases respectively every year. As part of the mass treatment activities conducted door-to-door in all regions of the country, Community Health Workers were trained to identify suspected cases of trichiasis. The suspected cases identified were examined for diagnostic confirmation to estimate the prevalence of TT.

**Tropical Data Survey: Methodology recommended by WHO.** The results of surveys from 2006, 2009, 2011; data from mass treatment, routine data in health facilities from 2014 to 2016 and TT case data from the project from June 2015 to December 2016 were collated. Taking all these data into account, 07 health districts from three regions were selected. These are the health districts of Est-Mono and Anié in the Plateaux region; Tchaoudjo and Tchamba in the Central region; Assoli, Dankpen and Kéran in the Kara region. The aim of this survey was to determine the extent of trachoma among children aged 1 to 9 years on the one hand, and on the other hand, to estimate the prevalence of TT in people aged 15 years and over according to the Tropical Data protocol recommended by WHO. The Tropical Data protocol [7] protocol is a document that outlines important design principles for trachoma prevalence surveys following interventions to eliminate trachoma as a public health problem and WHO recommendations for implementation of these surveys.

## 3. Results

### 3.1 Prevalence of TF and TT in the 2006, 2009 and 2011 surveys

**2006 Survey.** In 2006, a prevalence of 60 to 80% of TF among children aged 6 to 9 years was found in the Binah district (Kara Region).

**2009 Survey.** In 2009, the survey revealed a higher prevalence of TF in the Central region (5.7%) compared to the other regions (3.5% in the Kara region and 2.6% in the Savane region). The highest prevalences of TF were recorded respectively in Sotouboua (11.3%) and Blitta (10.9%) districts. Regarding TT, only the Kara region recorded 2 cases in the Bassar and Doufelgou districts. Table 1 summarizes the results of the 2009 survey.

**2011 Survey.** The 2011 survey in the three districts (Blitta, Sotouboua and Binah) in which TF appeared to be prevalent at levels above the elimination threshold reported very low prevalences ranging from 0 (Sotouboua district) to 0.2% (Blitta district) in the Central region

**Table 1. Distribution of TF among children aged 1 to 9 years in the three regions of Northern Togo, 2009 survey.**

| REGIONS | DISTRICTS | TRACHOMA | | | |
|---|---|---|---|---|---|
| | | Total number examined | Number of TF | Percentage of TF (%) | Number of TT |
| CENTRALE | BLITTA | 1900 | 208 | 10.9 | 0 |
| | TCHAOUDJO | 3200 | 34 | 1.1 | 0 |
| | TCHAMBA | 1300 | 23 | 1.8 | 0 |
| | SOTOUBOUA | 1800 | 204 | 11.3 | 0 |
| **Total 1** | | **8200** | **469** | **5.7** | **0** |
| KARA | ASSOLI | 600 | 8 | 1.3 | 0 |
| | BASSAR | 1700 | 55 | 3.2 | 1 |
| | DANKPEN | 1300 | 32 | 2.5 | 0 |
| | DOUFELGOU | 1900 | 109 | 5.7 | 1 |
| | KERAN | 1200 | 60 | 5.0 | 0 |
| | KOZAH | 3400 | 92 | 2.7 | 0 |
| **Total 2** | | 10 100 | 356 | 3.5 | 2 |
| SAVANES | KPENDJAL | 1200 | 17 | 1.4 | 0 |
| | OTI | 1600 | 10 | 0.6 | 0 |
| | TANDJOARE | 1500 | 7 | 0.5 | 0 |
| | TONE | 2400 | 140 | 5.8 | 0 |
| **Total 3** | | **6700** | **174** | **2.6** | **0** |

TF = trachomatous inflammation—follicular TT = Trachomatous Trichiasis

**Table 2. Distribution of TT among children aged 1–9 years and women aged 15 years and older in three districts of Northern Togo, 2011 Survey.**

| Region | District | Number of the villages | Children from 1 to 9 years old | | | Women aged 15 years and over | | |
|---|---|---|---|---|---|---|---|---|
| | | | Number of children examined | Prevalence TF | Prevalence TF (%) | Number of women examined | Prevalence TT | Prevalence TT (%) |
| Centrale | Blitta | 20 | 1400 | 2 | 0,1 | 1000 | 1 | 0,1 |
| | Sotouboua | 20 | 1400 | 0 | 0 | 1000 | 1 | 0,1 |
| Kara | Binah | 20 | 1400 | 2 | 0,1 | 1000 | 7 | 0,7 |

TF = trachomatous inflammation—follicular TT = Trachomatous Trichiasis

and 0.1% for Binah district in the Kara region (Table 2). Among women over 15 years of age, the prevalence of TT was 0.7% in the Binah district and 0.1% in the other two districts.

**Active search for TT cases during mass treatment.** Community Health Workers reported 5665 suspected TT cases during the 2015 mass treatment. Only 66% of the suspected cases were examined for which 0.79% (17 cases) were confirmed as TT cases including 10 confirmed cases in Savanes region and 7 confirmed cases in Kara region.

At the end of the 2017 mass treatment, Community Health Workers had notified 588 cases of eyelashes rubbing on the eyes including 13 confirmed cases of TT by senior ophthalmic technicians in 5 health regions (Savanes, Kara, Centrale, Plateaux and Maritime) resulting in a TT prevalence of 2.2%.

**TT screening and surgery project from June 2015 to December 2016.** The project diagnosed 108 people with TT with 159 eyes to cure. 116 eyes received surgery while 43 eyes received epilation of eyelashes for those who refused surgery.

### 3.2 Tropical data survey

The survey carried out in the 7 districts according to the Tropical Data protocol recommended by the WHO revealed a prevalence of TF of less than 1% in all districts among children aged 1 to 9 years and a very low prevalence of TT among 15-year-olds (Table 3).

At the end of the survey, 27 patients were identified in need for surgery. 19 of them were surgered while 6 refused surgery and therefore benefited from epilation of eyelashes. In Tchamba district, one (01) patient diagnosed had died before the surgery period and one (01) other person was absent in Anié district during the surgery period.

**Table 3. Distribution of TF among children aged 1–9 years and TT among those over 15 years in 2017.**

| Region | District | Number of villages | Number of households | Children from 1 to 9 years old | | | Adults over 15 years of age | | |
|---|---|---|---|---|---|---|---|---|---|
| | | | | Number of children examined | Number of TF | Age-adjusted TF prevalence (%) | Number >15 years examined | Number of TT | TT prevalence (%) adjusted for gender and age |
| Kara | Kéran | 25 | 750 | 1511 | 6 | 0,38 | 1573 | 1 | 0,03 |
| | Assoli | 25 | 750 | 1479 | 5 | 0,30 | 1849 | 0 | 0,00 |
| | Dankpen | 25 | 750 | 1955 | 8 | 0,44 | 2079 | 4 | 0,06 |
| Centrale | Tchaoudjo | 25 | 750 | 1478 | 11 | 0,45 | 1574 | 7 | 0,13 |
| | Tchamba | 25 | 750 | 1522 | 7 | 0,56 | 1970 | 6 | 0,14 |
| Plateaux | Anié | 25 | 750 | 1414 | 4 | 0,27 | 1329 | 0 | 0,00 |
| | Est-Mono | 25 | 750 | 1210 | 6 | 0,33 | 1606 | 0 | 0,00 |

TF = trachomatous inflammation—follicular TT = Trachomatous Trichiasis

## 4. Discussion

### 4.1 Summary of main results

The analysis of the fight against trachoma since its launch in 2006 shows the abnegation of the program to eradicate this disease from the country. Indeed, the results show:

- In 2006, a prevalence of 60 to 80% of TF among children aged 6 to 9 years was found in the Binah district (Kara Region);

- In 2009, TF prevalence was 5.72% in the Central Region and 2 cases of trichomatous trichiasis (TT) in the Bassar and Doufelgou districts (Kara Region). Among women over 15 years of age, the overall prevalence of TT was 0.7%;

- In 2011, TF prevalence among children aged 1 to 6 years, less than 1% in the districts of Sotouboua and Blitta (Central region) and the district of Binah (Kara region);

- 0.79% of investigated suspected TT cases were confirmed in 2015 CTs by CSAs;

- At the end of the 2017 mass treatment, 13 confirmed cases of TT by senior ophthalmic technicians were recorded in 5 health regions (Savanes, Kara, Centrale, Plateaux and Maritime) giving a prevalence of TT of 2.21%;

- The TT screening and surgery project resulted in the diagnosis of 108 people with TT with 159 eyes of which 116 eyes received surgery and 43 eyes received epilation of eyelashes;

- From 2014 to 2017, eye care centers served 2,606,978 people over 15 years of age in whom no TF cases were recorded but 199 TT cases were routinely diagnosed;

- The 2017 survey conducted in the 7 districts identified a TF prevalence of less than 1% all districts combined among children aged 1–9 years and a TT prevalence also very low in adults over 15 years;

- From 2018 to 2020, surgery was practiced on 24 TT cases and 19 eyes were waxed.

### 4.2 Involvement of health authorities and partners

The elimination of a neglected tropical disease such as trachoma in a country is the result of a combination of multilevel efforts, involvement of multiple stakeholders in close collaboration with affected communities, national mobilization of funding to implement and sustain interventions, and cross-sectoral efforts to improve the living conditions of the people [8]. Political will and the support of various partners were the key elements that made this achievement.

**4.2.1 Political commitment.** Political commitment was reflected early in the fight with the establishment of the the National Program for Blindness control in 1989. The Surgery, Antibiotics, Facial cleanliness and Environmental modification (SAFE) strategy recommended by WHO for the elimination of trachoma [9,10] and adopted in several countries [10,11] was the one implemented by the Togolese authorities through the National Program for Blindness control.

Patients identified with TT (through surveys, routine consultations in eye care centers, mobile eye screening and management campaigns, during advanced strategies used by some eye care centers, as well as during door-to-door visits during the integrated mass treatment campaigns against onchocerciasis, schistosomiasis and geo-helminthiasis in 2015 and 2017) and for whom TT surgery was necessary, were managed. Patients with TT who present to health centers where eye care services are available are operated on in fixed posts and those screened in advanced strategies are managed in the health facility closest to the patient.

Trichiasis surgery is performed according to the standard procedure of the WHO document. After surgery, each patient received a tube of tetracycline ointment for application twice a day. The senior ophthalmic technicians who did the TT surgery were supervised by the ophthalmologists during the surgery. The operated patients return the day after the surgery for the removal of the dressing and an examination of the wound by the senior ophthalmic technicians who performed the surgery. 100% of the operated patients are reviewed on the eighth (8$^{ème}$) day after the surgery for a check-up of the wound and a removal of the non-absorbable sutures if necessary. For the 3 to 6 month postoperative follow-up, most of patients did not respond to the follow-up appointment. Therefore, the Program did not have the opportunity to perform long-term follow-up for most of the patients.

Facial cleansing activities were highly integrated with those of the integrated onchocerciasis, schistosomiasis and geo-helminthiasis mass treatment, as well as with those of the Hygiene and Sanitation service. Each year, during the implementation of the integrated mass treatment for onchocerciasis, schistosomiasis and geo-helminthiasis, which is carried out through a door-to-door approach in each region, the Community Health Workers sensitize household members on hand and face washing. Administrative and religious authorities and teachers were also involved through awareness raising activities.

The environmental health hazards to which the country has been exposed for several decades are open defecation, inadequate or inappropriate disposal of medical and garbage-like wastes, unsafe water and food, lack of or insecure vector control, and chemical exposure. Many strategies have been developed to reduce environmental pollution.

The ingenious idea of merging all the NTD programmes into a single programme has been key in this fight. Thanks to this decision it was possible to integrate screening and sensitization activities with the mass treatment activities. As part of the door-to-door mass treatment activities in all regions of the country, Community Health Workers were trained to identify suspected cases of trichiasis (deformed upper eyelids/presence of eyelashes touching the front of the eye/notification of eyelashes rubbing on the eyes). This door-to-door strategy provided a national overview of the situation, particularly in the Plateaux (Akébou, Wawa, Amou, Danyi, Kpélé, Kloto, Ogou, Moyen Mono and Haho) and Maritime (Yoto, Avé, Zio, Lacs, Vo and Bas- Mono) districts, which had not benefited from a prevalence survey for active or sequelae trachoma.

**4.2.2 Partnership.** The extensive interventions have been successfully implemented with financial and technical support from partners. The program has established a strong partnership with international organizations such as WHO, Center for Disease Control and Prevention (CDC) Atlanta, Sightsavers, Tropical Data and Health and Development International (HDI) with funding from the Bill and Melinda Gates Foundation and United States Agency for International Development (USAID) through Family Health International 360 (FHI 360).

## 4.3 Defaults

Breaches have been observed and this could be the result of a long process of elimination. Changes in health authorities, system reforms and changes in public health priorities, among others, are factors that could affect the continuity of public health interventions [8].

We noted that a false start was made in the evaluation of the epidemiological situation of trachoma in Togo. Indeed, the "integrated threshold mapping" of 2006 in the Binah district in the Kara region and of 2009 in 14 health districts out of the 15 in the three northern regions of Togo (Centrale, Kara and Savanes) with the exception of the Binah district was alarming in revealing TF prevalences that exceeded the elimination threshold for active trachoma. These results were probably due to overdiagnosis, with the presence of a high number of follicles of

any size and in any conjunctival location being erroneously considered as indicative of TF, as explicitly acknowledged in the 2009 publication [9]. The 2011 results, five (05) years after the 2006 results and two years after the 2009 results despite the absence of mass treatment, confirm the overdiagnosis hypothesis. The more rigorous 2011 surveys found near absence of active trachoma in all three districts especially in the districts where TF prevalences were highest.

Methodological weaknesses related to the indicators to be collected, were found in the surveys run between 2006 and 2016. For example, TT was not searched for in the first two. For the 2006, 2009 and 2011 surveys, TT cases were not searched for. These shortcomings were corrected in the continuation of the surveys conducted in 2017 and allowed the best possible identification of all stages of trachoma (TF and TT) in order to propose strategies to achieve the elimination of trachoma as a public health problem in 2022. A clearer definition of the key indicators to follow-up from the beginning of the monitoring process would have allowed Togo to eliminate trachoma a few years earlier.

## 4.4 Follow-up and post-validation monitoring

In 2011, Togo, like other member countries of the WHO African Region, adopted the Integrated Disease Surveillance and Response technical guide which aims to strengthen the national surveillance system. Trachoma is included in the Integrated Disease Surveillance and Response technical guide and is one of the notifiable diseases and events. Outside of the Integrated Disease Surveillance and Response technical guide, routine data on trachoma are collected by the ophthalmic technicians and the NTD focal points.

NCD focal points are trained on how to enter trachoma surveillance indicators into District Health Information Software 2 (DHIS2). The indicators on TF and TT are integrated into the DHIS2 and filled in by the NTD district focal points and the ophthalmic technicians in the districts. Routine TF and TT cases are reported by the senior ophthalmic technicians and entered into the DHIS2 by the NTD district focal points. Awareness raising on hand and face washing is performed every year from door to door by the Community Health Workers during the integrated mass treatment which takes place in all the regions of Togo and by the managers of health facilities during routine activities in fixed or advanced posts. The maintenance of post-elimination activities, monitoring and evaluation integrated into national programs has been the key to success in countries where trachoma has been eliminated such as Oman, Morocco and Iran [12].

## 4.5 Elimination of trachoma in Togo

The 2017 survey, conducted according to the WHO recommended methodology, revealed that the prevalence of key indicators was below the WHO elimination threshold for trachoma. The surveillance system set in place collected data on routine service delivery for TT in the country indicating both the existence of a system for identification and management of incident cases of TT. Based on such data, a country case was prepared and submitted to WHO. The application provided sufficient evidence that Togo has met the criteria established to validate the elimination of trachoma as a public health problem which are [3]:

a. the prevalence of TF among children aged 1–9 years is <5% in all nationally defined administrative units suspected of having trachoma;

b. the prevalence of trichiasis among adults aged 15 years and over is less than 0.2% in all administrative units previously endemic for trachoma;

c. evidence of capacity to manage the incident of emerging trichiasis in the communities.

An independent regional dossier review panel convened by WHO reviewed the case and confirmed that the criteria for elimination were met. Based on this evidence, WHO validated and formally recognized that Togo had eliminated trachoma as a public health problem by May 2022 [4].

## 5 Conclusion

Thanks to a synergy of efforts by the Togolese government, governmental and non- governmental partners over the past 15 years, Togo becomes the fourth country in Africa to be certified by the World Health Organization (WHO) as having reached the elimination milestone after Morocco in 2016, Ghana in 2018, Gambia in 2021 and Malawi in 2022. By reaching this milestone, Togo has improved the quality of life of people living in former trachoma endemic areas by avoiding visual impairment and eventual blindness due to this disease. Post- validation monitoring and surveillance is essential to detect the resurgence of the disease because elimination as a public health problem is a reversible state.

## Acknowledgments

We are grateful to the National Programme for the Control of Neglected Tropical Diseases for providing us with the necessary documents and to the German Leprosy and Tuberculosis Relief Association for its technical support in writing this article.

## Author Contributions

**Validation:** Didier Koumavi Ekouevi.

**Writing – original draft:** Akila Wimima Bakoubayi.

**Writing – review & editing:** Akila Wimima Bakoubayi, Denis Agbenyigan Yawovi Gadah, Piham Gnossike, Wendpouiré Ida C. Zida-Compaore, P'tanam P'kontème Bando, Kamevor Alaglo, Maweke Tchalim, P'niwè Patchali, Alexandra Bitty-Anderson.

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
