## [Decision Letter · Decision Letter 0]

3 Feb 2023

Dear Dr Akila Wimima,

Thank you very much for submitting your manuscript "How did Togo, a West African country, eliminate lymphatic filariasis, dracunculiasis, human African trypanosomiasis and trachoma as a public health problem in only 11 years? Sharing experience: The case of trachoma" for consideration at PLOS Neglected Tropical Diseases. As with all papers reviewed by the journal, your manuscript was reviewed by members of the editorial board and by several independent reviewers. In light of the reviews (below this email), we would like to invite the resubmission of a significantly-revised version that takes into account the reviewers' comments. 

Editorial comments

Title of paper since this manuscript focuses on trachoma, you might want to review the title to be more generic on the 4 eliminated NTDs but specific on trachoma, for example: “Togo’s success story in eliminating four NTDs as a public health problem within a decade: Lessons learnt from trachoma elimination"

We cannot make any decision about publication until we have seen the revised manuscript and your response to the reviewers' comments. Your revised manuscript is also likely to be sent to reviewers for further evaluation.

Sincerely,

Jeremiah M. Ngondi, MB.ChB, MPhil, MFPH, Ph.D

Academic Editor

Elsio Wunder Jr

Section Editor

Editorial comments

Title of paper requires revision since this manuscript focuses on trachoma. You might want to review the title to be more generic on the 4 eliminated NTDs but specific on trachoma, for example: “Togo’s success story in eliminating four NTDs as a public health problem within a decade: Lessons learnt from trachoma elimination"

I note that the authors are primarily French speakers thus some context has possibly been lost in the English translations. As highlighted by the reviewers, I suggest that the manuscript be copy edited to correct the language and also include the appropriate English acronyms. 

Reviewer's Responses to Questions

**Key Review Criteria Required for Acceptance?**

**Methods**

-Are the objectives of the study clearly articulated with a clear testable hypothesis stated?

-Is the study design appropriate to address the stated objectives?

-Is the population clearly described and appropriate for the hypothesis being tested?

-Is the sample size sufficient to ensure adequate power to address the hypothesis being tested?

-Were correct statistical analysis used to support conclusions?

-Are there concerns about ethical or regulatory requirements being met?

Reviewer #1: This article is relevant and could be a good reference for countries that are still struggling to achieve trachoma elimination as a public health problem. However, the paper is not very well written and lacks coherence and smooth flow of the historical development of the program. From the outset, the title of the paper doesn't appear to be appropriate. The article is all about trachoma elimination in Togo but not clear why it's titled as such. The authors can discuss the success stories of other programs in the introduction or in the Discussion sections but the title of the paper should only focused on trachoma elimination in Togo. 

One other general comment is that the paper has to be rewritten by a native English language speaker. There are some accronyms that I suspect have come from a French language. Examples are; CT, PNMTN, MSHPAUS, PNLC, MTD, TDM, FT, ASC, CHANCE, OSI, TSO, ICMR, IRMS, NCDFPs, OSI, etc. These should be replaced by the appropriate English accronyms.

The baseline trachoma prevalence data were not approrpriately presented. On page 7, it says the prevalence of trachoma in the Binah district, Kara region. was 60 - 80%. Looking at the following results of the survey in 2009, which was on average less than 10%, the prevalence reported for Binah district doesn't seem accurate. There must have been a methodological error introduced by the Integrated Threshold Mapping approach. There is no detailed summary table provided for the results of the 2006 survey. That should be corrected.

There are lots of omissions or wrongly coined or incomplete statements. For example, in many places, it says "WHO awarded Togo for becoming ...." Awarded what? 

In relation to TT management, the phrase "eyelash removal" has been repeatedly used. This has to be replaced by the correct term "epilation". On Table 3, the heading has to change; "Results of TFs and TTs" is not appropriate use of the terms.

"CHANCE" should be replaced by "SAFE" strategy. 

On page 15, An independent "case review panel" should be replaced by "Dossier review panel".

Reviewer #2: The objectives of the study are clearly stipulated, and the article follows standard trachoma methodology guidelines.

No issues

Reviewer #3: This article is based mainly on events that took place under the trachoma elimination program in Togo so even though it has clear objectives, no hypothesis is stated and cannot be assessed against a study design, sample sizes and methods and did not involve any statistical analysis. it is just a review of processes leading to the attainment of trachoma elimination together with 3 other diseases within 11 years.

**Results**

-Does the analysis presented match the analysis plan?

-Are the results clearly and completely presented?

-Are the figures (Tables, Images) of sufficient quality for clarity?

Reviewer #1: I have discussed some of the Results section already in the above section. It requires rewriting.

Reviewer #2: results are descriptive and clear

Reviewer #3: The article involved the presentation of already analysed survey data and presentation in a very simple form to indicate how decisions toward elimination and certification were taken. 

In my opinion the content of the paper does not adequate justify the topic and so the topic or the content needs to be reviewed to fall in alignment. The content only focuses on trachoma elimination with little mention of the other three diseases.

**Conclusions**

-Are the conclusions supported by the data presented?

-Are the limitations of analysis clearly described?

-Do the authors discuss how these data can be helpful to advance our understanding of the topic under study?

-Is public health relevance addressed?

Reviewer #1: The conclusion is ok but but needs rewriting. There shouldn't be quoted statements in the conclusion.

Reviewer #2: conclusion is well supported by findings

the article has public health relevance for neglected tropical diseases

Reviewer #3: The conclusions are supported by the presented data, however the limitations of the process and events towards these achievements are not adequately highlighted in the article.

**Editorial and Data Presentation Modifications?**

Reviewer #1: Major revision required.

Reviewer #2: Minor comments:

The article has several typos that need to be corrected. 

Author summary 

Line 47. Type whichwas to be which was

Line 49 type screenand

Line 50 should be achieved in all

Table 1has a larger different font and format . please revise 

Line 186 -191; can SAFE be used instead of CHANCE as the article in is English and not French?

Reviewer #3: On page 10, line 116, the heading “Detection of suspected TT cases during CT scans from 2015 to 2017” needs to be reviewed and clarified since CT in the manuscript apparently stands for community treatment and not computer tomography as it appears here. 

There are a few other editorial errors that require attention.

**Summary and General Comments**

Reviewer #1: The article doesn't merit publication in its current form.

Reviewer #2: This is a god article and will be useful for other countries nearing elimination. it should be noted that additional countries like Malawi have also declared Trachoma elimination in Sept 2022, so it may need updating the figures 

Minor comments:

The article has several typos that need to be corrected. 

Author summary 

Line 47. Type whichwas to be which was

Line 49 type screenand

Line 50 should be achieved in all

Table 1has a larger different font and format . please revise 

Line 186 -191; can SAFE be used instead of CHANCE as the article in is English and not French?

Reviewer #3: Review Comments: How did Togo, a West African country, eliminate lymphatic filariasis, dracunculiasis, human African trypanosomiasis, and trachoma as a public health problem in only 11 years? Sharing experience: The case of trachoma

This is a great paper about an achievement or feat worth celebrating in global health. Even though a country’s achievements, various partners collaboratively contributed technically and financially to make this a reality with the government being the centerpiece in driving these efforts with insightful lessons for other disease programs in Togo and other sub-Saharan African countries. The paper acknowledges the different roles played by the different organizations and civil society with the WHO being a significant partner in the scheme of affairs. The role of government in the scheme of affairs does not come out clearly and strongly to me.

There have been several technical opportunities, in this journey, together with challenges, and learning to drive other programs within Togo and for other countries in sub-Saharan Africa and elsewhere globally. 

In my reading of the manuscript, I realized there were no surveys to validate the data already collected during surveys conducted in 2006, 2009, and 2011.

Mapping data was also not included in the write-up and how it was used in planning treatment to enable follow-up surveys and demonstration of the trajectory of transmission over the period. It is however clear that adequate baseline data was collected to guide the program. 

Various methods were adapted for the surveys undertaken in 2006, 2009, 2011, and 2016. The guidelines applied and some analysis of the survey methodologies to bring out differences that could have impacted the results obtained and recommended methods moving forward would have been insightful and informative.

Readers would be interested to know what happened between the period from 1989, with the establishment of the NCCP till 2016 when the mapping or baseline surveys were conducted which is also unclear in the manuscript. 

On page 10, line 116, the heading “Detection of suspected TT cases during CT scans from 2015 to 2017” needs to be reviewed and clarified since CT in the manuscript apparently stands for community treatment and not computer tomography as it appears here. 

There are a few other editorial errors that require attention.

In all this is an interesting and insightful paper on an important and critical subject in the history of the Global NTD Program with great lessons to inform many public health programs in sub-Saharan Africa. 

I recommend this article for publication after addressing the comments from the review. 

Thank you for the opportunity to review this manuscript.

PLOS authors have the option to publish the peer review history of their article (what does this mean?). If published, this will include your full peer review and any attached files.

Reviewer #1: No

Reviewer #2: Yes: Dr Khumbo Kalua

Reviewer #3: Yes: Nana-Kwadwo Biritwum
---

## [Decision Letter · Decision Letter 1]

13 Apr 2023

Dear Dr BAKOUBAYI,

Thank you very much for submitting your manuscript "The elimination of trachoma as a public health problem in Togo: Successes and challenges" for consideration at PLOS Neglected Tropical Diseases. As with all papers reviewed by the journal, your manuscript was reviewed by members of the editorial board and by several independent reviewers. The reviewers appreciated the attention to an important topic. Based on the reviews, we are likely to accept this manuscript for publication, providing that you modify the manuscript according to the review recommendations. 

Sincerely,

Jeremiah M. Ngondi, MB.ChB, MPhil, MFPH, Ph.D

Academic Editor

Elsio Wunder Jr

Section Editor

Reviewer's Responses to Questions

**Key Review Criteria Required for Acceptance?**

**Methods**

-Are the objectives of the study clearly articulated with a clear testable hypothesis stated?

-Is the study design appropriate to address the stated objectives?

-Is the population clearly described and appropriate for the hypothesis being tested?

-Is the sample size sufficient to ensure adequate power to address the hypothesis being tested?

-Were correct statistical analysis used to support conclusions?

-Are there concerns about ethical or regulatory requirements being met?

Reviewer #2: (No Response)

Reviewer #3: The objectives are clearly stated but because it's a review paper there is no stated hypothesis. The endemic populations are well described together with the studies or operational research activities that were undertaken there.

**Results**

-Does the analysis presented match the analysis plan?

-Are the results clearly and completely presented?

-Are the figures (Tables, Images) of sufficient quality for clarity?

Reviewer #2: (No Response)

Reviewer #3: The analysis of the data which is mainly retrospective has been well done and presented. The data employed for the drafting of the manuscript was basically longitudinal retrospective programmatic review data already analyzed and applied for programmatic decision making which is in order. This data has been presented in a logical chronological manner which should make a lot of sense to any reader. The tables used are of fair quality and provide a lot of clarity to the data analyzed and presented.

**Conclusions**

-Are the conclusions supported by the data presented?

-Are the limitations of analysis clearly described?

-Do the authors discuss how these data can be helpful to advance our understanding of the topic under study?

-Is public health relevance addressed?

Reviewer #2: (No Response)

Reviewer #3: Being a review paper, the authors are limited in identifying limitations but the main limitation as I see it is the use of retrospective programme data the authors have limited control over at the time of writing up.

**Editorial and Data Presentation Modifications?**

Reviewer #2: (No Response)

Reviewer #3: The revised version of the paper as it stands now is a marked improvement of the original version. It is well-written, flows well and requires veery little editorial work.

**Summary and General Comments**

Reviewer #2: Review 

Overall most concerns have been addressed. However the paper still has a lot of typos and need a copy editor.

Abstract:

Line 22 ADD dossier to state Country’s dossier submission

Line 25. Separate alsoincluded to also included 

Line 27 . remove has .

All results should be stated in the past tense.

Line 28 should read was < than 5% and line 30 was less than 0.2% 

Sentence 37 starting with BUT . please rephrase

Author Summary 

Line 47 . screen and manage . there is typo

Introduction

Line 71 typo whichwas

Line 72 five to be replaced by fifth

Line 73 Malawi was declared in September after Togo . please change

Methods

line 86. We did not require consent . if this true? Even if this was secondary data, consent /permission from Ministry of Health to use it was needed . please clarify that you did not obtain consent 

Line 102 . give a reference

Line 113 - was this the WHO standardized recognized methodology? Please clarify the methodology 

Did the survey also address the issue of TT ? this is not mentioned?

Line 119 : typo resultsof

Line 132 : collated ? please explain ?

Line 133 : typo -allthese

It is not clear when the surveys with Tropical data were conducted . is it after 2016 ? please clarify ?

Line 172 is not clear. What is 159 eyes to cure ? 

Line 179 were surgered ? not correct English 

Line 227 againstonchocerciasis – typo

Line 230 – typo advance strategies

Line 332. Please check . I think Togo came before Malawi

Reviewer #3: Thank you for the opportunity to review a revised version of the article “The elimination of trachoma as a public health problem in Togo: Successes and Challenges”. This is a marked improvement on the previous manuscript and broadly addresses my concerns and comments on the previous version. The title better matches the content of the manuscript. The write-up clearly distinguishes the data submitted to the WHO for certification of trachoma elimination from post-elimination surveillance data collected from the surveillance systems that are in place in a logical and chronological order making the paper very easy to follow and make sense of. The flow of the write-up is now commendable but needs slight editing. 

The objective of this review paper is clear with great lessons for other programmes regarding the successes and challenges of the process. The introduction contains adequate background information on the global trachoma elimination programme and the strides made on the African continent. 

The programme’s benefit from the broader government policies, strategies, processes, and resources is commendable and well documented. It presents great lessons for countries to learn from in presenting their programmes to governments for support. 

At this point, it is clear that the paper is very useful and relevant to the current strategies of the WHO NTD 2030 Roadmap and global trachoma elimination. The lessons presented will be of immense assistance to other programmes, especially in sub-Saharan Africa.

I recommend that paper for publication after the required editorial corrections.

PLOS authors have the option to publish the peer review history of their article (what does this mean?). If published, this will include your full peer review and any attached files.

Reviewer #2: Yes: Dr Khumbo Kalua

Reviewer #3: Yes: Nana-Kwadwo Biritwum

Figure Files:

Data Requirements:

Reproducibility:

References

---

## [Editor Report · Decision Letter 2]

6 Jun 2023

Dear Dr BAKOUBAYI,

We are pleased to inform you that your manuscript 'The elimination of trachoma as a public health problem in Togo: Successes and challenges' has been provisionally accepted for publication in PLOS Neglected Tropical Diseases.

Best regards,

Jeremiah M. Ngondi, MB.ChB, MPhil, MFPH, Ph.D

Academic Editor

Elsio Wunder Jr

Section Editor

---

## [Editor Report · Acceptance letter]

4 Jul 2023

Dear Dr BAKOUBAYI,

We are delighted to inform you that your manuscript, "The elimination of trachoma as a public health problem in Togo: Successes and challenges," has been formally accepted for publication in PLOS Neglected Tropical Diseases.

Best regards,

Shaden Kamhawi

co-Editor-in-Chief

Paul Brindley

co-Editor-in-Chief
